# Prognostic Inflammatory Index Based on Preoperative Peripheral Blood for Predicting the Prognosis of Colorectal Cancer Patients

**DOI:** 10.3390/cancers13010003

**Published:** 2020-12-22

**Authors:** Jinming Fu, Ji Zhu, Fenqi Du, Lijie Zhang, Dapeng Li, Hao Huang, Tian Tian, Yupeng Liu, Lei Zhang, Ying Liu, Yuanyuan Zhang, Jing Xu, Shuhan Meng, Chenyang Jia, Simin Sun, Xue Li, Liyuan Zhao, Ding Zhang, Lixin Kang, Lijing Gao, Ting Zheng, Sanjun Cai, Yanlong Liu, Yashuang Zhao

**Affiliations:** 1Department of Epidemiology, College of Public Health, Harbin Medical University, Harbin 150081, China; Fu_jinming@hrbmu.edu.cn (J.F.); ldroc1987@gmail.com (D.L.); huanghao@hrbmu.edu.cn (H.H.); 102415@hrbmu.edu.cn (T.T.); liuyupeng@wmu.edu.cn (Y.L.); zhanglie@hrbmu.edu.cn (L.Z.); yingliu0531@gmail.com (Y.L.); zhangyuanyuan941012@gmail.com (Y.Z.); xu806663632@gmail.com (J.X.); mengsh@hrbmu.edu.cn (S.M.); jiachenyang1126@hrbmu.edu.cn (C.J.); sunsimin@hrbmu.edu.cn (S.S.); 2018020158@hrbmu.edu.cn (X.L.); 2018020181@hrbmu.edu.cn (L.Z.); ZhangD_95@hrbmu.edu.cn (D.Z.); kanglixin@hrbmu.edu.cn (L.K.); 2020020185@hrbmu.edu.cn (L.G.); zitty@hrbmu.edu.cn (T.Z.); 2Department of Radiation Oncology, Fudan University Shanghai Cancer Center, Shanghai 200032, China; leoon.zhu@gmail.com (J.Z.); ljzhang15@fudan.edu.cn (L.Z.); 3Department of Colorectal Surgery, Harbin Medical University Cancer Hospital, Harbin Medical University, Harbin 150081, China; 2019021558@hrbmu.edu.cn; 4Department of Colorectal Surgery, Fudan University Shanghai Cancer Center, Shanghai 200032, China; caisanjun@gmail.com

**Keywords:** colorectal cancer, peripheral blood cell, prognostic inflammatory index, prognosis, nomograms

## Abstract

**Simple Summary:**

Inflammation plays a critical role in the progression of colorectal cancer (CRC). Peripheral blood cell counts could reflect the extent of systemic inflammation and are readily available in clinical practice. The aim of our study was to construct a novel prognostic inflammatory index (PII) by integrating the blood cell counts associated with prognosis and to evaluate and validate the prognostic value of PII in two independent CRC cohorts. Multivariate Cox analyses in the training cohort of 4154 CRC patients indicated that high OS-PII (>4.27) and high DFS-PII (>4.47) were significantly associated with worse OS (HR: 1.330, *p* < 0.001) and worse DFS (HR: 1.366, *p* < 0.001), which has been validated in the external validation cohort of 5161 patients. Both OS-PII and DFS-PII have a stable prognostic performance at various follow-up times, and the nomograms based on OS-PII and DFS-PII achieved good accuracy in personalized survival prediction of patients with CRC.

**Abstract:**

Host inflammation is a critical component of tumor progression and its status can be indicated by peripheral blood cell counts. We aimed to construct a comprehensively prognostic inflammatory index (PII) based on preoperative peripheral blood cell counts and further evaluate its prognostic value for patients with colorectal cancer (CRC). A total of 9315 patients with stage II and III CRC from training and external validation cohorts were included. The PII was constructed by integrating all the peripheral blood cell counts associated with prognosis in the training cohort. Cox analyses were performed to evaluate the association between PII and overall survival (OS) and disease-free survival (DFS). In the training cohort, multivariate Cox analyses indicated that high OS-PII (>4.27) was significantly associated with worse OS (HR: 1.330, 95% CI: 1.189–1.489, *p* < 0.001); and high DFS-PII (>4.47) was significantly associated with worse DFS (HR: 1.366, 95% CI: 1.206–1.548, *p* < 0.001). The prognostic values of both OS-PII and DFS-PII were validated in the external validation cohort. The nomograms achieved good accuracy in predicting both OS and DFS. Time-dependent ROC analyses showed that both OS-PII and DFS-PII have a stable prognostic performance at various follow-up times. The prognostic value of tumor-node-metastasis staging could be enhanced by combining it with either OS-PII or DFS-PII. We demonstrated that PIIs are independent prognostic predictors for CRC patients, and the nomograms based on PIIs can be recommended for personalized survival prediction of patients with CRC.

## 1. Introduction

Globally, colorectal cancer (CRC) ranks second in terms of mortality, with an estimated 881,000 deaths in 2018 [1]. In China, although CRC is the fifth leading cause of cancer death among both men and women, the mortality of CRC has been increasing in the recent decade [2,3]. Radical resection is the most common treatment for CRC patients; however, approximately one-half of patients will experience a recurrence within the first 3 years after surgery [4]. Tumor-node-metastasis (TNM) staging and specific histological features have been identified as prognostic factors for CRC [5,6,7]. However, patients with equivalent characteristics may have different survival. Various molecular biomarkers have been reported to be associated with clinical outcomes [8,9,10], but high cost and sophisticated laboratory measurement limit the application in routine clinical practice. These have led to intense interest in exploring new, cheap, and convenient prognostic biomarkers.

It is now recognized that inflammation is a hallmark of cancer and is closely related to tumor progression [11,12,13,14]. Host inflammation response can suppress the antitumor function of adaptive immunity and break the balance between the immune system and malignant tumors, thereby causing the poor prognosis of patients [15]. The inflammatory process frequently causes changes in numerous hematological parameters, such as peripheral blood cell counts and levels of C-reactive protein and albumin. In comparison, peripheral blood cell counts are easy to measure, inexpensive, and widely available in routine clinical practice.

Inflammatory markers, including neutrophil-to-lymphocyte ratio (NLR), platelet-to-lymphocyte ratio (PLR), lymphocyte-to-monocyte ratio (LMR), systemic immune-inflammation index (SII, platelets × neutrophils/lymphocytes), eosinophils and basophils, have been investigated for their prognostic roles in CRC patients [16,17,18,19,20,21,22,23,24,25]. However, these markers based on a single type or ratio of two or three types of blood cell counts failed to contain all the information of peripheral blood cells for prognosis prediction. Combining NLR, PLR and LMR may provide more information, but this ignores the fact that these markers share reiterative information (lymphocyte is part of both NLR and LMR).

Therefore, we separately assessed the relationship between all the six types of preoperative peripheral blood cell counts and the prognosis of CRC patients and constructed a prognostic inflammatory index (PII) by integrating the blood cell counts associated with prognosis. We evaluated the prognostic value of PII for CRC patients and validated it in an independent CRC cohort. Furthermore, we developed nomograms for personalized survival prediction of patients with CRC, which help clinicians to identify high-risk populations and develop treatment strategies.

## 2. Results

### 2.1. Patient Characteristics

A total of 9315 patients with primary CRC from two independent cohorts were included in this study, consisting of 4471 stage II and 4844 stage III patients. The median follow-up time was 70.0 months (interquartile ranges: 50.0–94.0) in the training cohort, with 1274 deaths during this period. The median follow-up time was 24.0 months (interquartile ranges: 12.0–41.0) in the external validation cohort, with 543 deaths during this period. Demographic and clinical characteristics of patients in the training cohort and external validation cohort are summarized in Table 1.

### 2.2. Identification of the PII and the Optimal Cut-Off Value

The results of restricted cubic spline (RCS) regression suggested that platelet, lymphocyte, and eosinophil counts have nonlinear relationships with overall survival (OS) (Appendix A), while platelet and eosinophil counts have nonlinear relationships with disease-free survival (DFS) (Appendix A). Thus, platelet, lymphocyte, and eosinophil counts were converted into binary variables for further Cox analyses (Appendix A). According to the results of univariate Cox analyses, platelet, lymphocyte, neutrophil, monocyte, and eosinophil counts were associated with OS, whereas only platelet, neutrophil, monocyte, and eosinophil counts were associated with DFS (Appendix A). As the types of blood cells associated with OS and DFS were different, we constructed OS-PII and DFS-PII separately.

The OS-PII was constructed using platelet, lymphocyte, neutrophil, monocyte and eosinophil counts with weights given by the corresponding coefficients from the multivariate Cox model (Appendix A): (1.878 × Platelet) + (1.370 × Lymphocyte) + (0.251 × Neutrophil) + (4.570 × Monocyte) + (2.094 × Eosinophil). The DFS-PII was constructed using platelet, neutrophil, monocyte, and eosinophil counts with weights given by the corresponding coefficients from the multivariate Cox model (Appendix A): (2.370 × Platelet) + (0.415 × Neutrophil) + (2.600 × Monocyte) + (2.437 × Eosinophil). X-tile 3.6.1 software was used to determine the optimal cut-off values for OS-PII and DFS-PII, which were 4.27 and 4.47, respectively (Appendix A). Patients were separated into low PII groups (OS-PII ≤ 4.27; DFS-PII ≤ 4.47) and high PII groups (OS-PII > 4.27; DFS-PII > 4.47) for further study.

The association of OS-PII and DFS-PII with clinicopathological characteristics within the training cohort is presented in Appendix A. OS-PII was associated with age, gender, body mass index (BMI), hypertension, tumor location, tumor diameter, differentiation degree, tumor invasion, and CA19-9 (*p* < 0.05). DFS-PII was associated with differentiation degree and CA19-9 (*p* < 0.05). In the validation cohort, OS-PII was associated with age, gender, tumor location, tumor diameter, differentiation degree, CEA, and CA19-9, while DFS-PII was associated with age, nerve invasion, vascular tumor thrombus, and CA19-9, which were consistent with the findings in the training cohort (Appendix A).

### 2.3. Prognostic Value of OS-PII and DFS-PII in the Training Cohort

As the Kaplan–Meier curves show in Figure 1, patients in the high OS-PII and high DFS-PII groups had significantly poorer survival (log-rank test, *p* ≤ 0.001). The 1-, 3-, 5- and 10-year OS and DFS rates of patients in the low OS-PII group were significantly higher than those of patients in the high OS-PII (Appendix A).

High OS-PII was significantly associated with worse OS in univariate Cox analysis (Table 2). Adjusting for age, gender, tumor location, tumor diameter, pathological classification, differentiation degree, histologic classification, TNM staging, nerve invasion, vascular tumor thrombus, CEA, CA19-9, postoperative chemotherapy and radiotherapy in the multivariate Cox analysis, OS-PII was still statistically associated with the OS of CRC (hazard ratio [HR]: 1.330, 95% confidence interval [CI]: 1.189–1.489, *p* < 0.001), which indicated that OS-PII was an independent prognostic predictor for CRC (Table 2).

High DFS-PII was significantly associated with worse DFS in univariate Cox analysis (Table 3). Upon multivariate Cox analysis, DFS-PII was also an independent prognostic predictor for patients with CRC (HR: 1.366, 95% CI: 1.206–1.548, *p* < 0.001) (Table 3).

### 2.4. Prognostic Value of OS-PII and DFS-PII in the Validation Cohort

The OS-PII and DFS-PII were further applied to the validation cohort to verify their transportability and generalizability. In the validation cohort, high OS-PII and high DFS-PII correlated significantly with worse OS (*p* < 0.001) and DFS (*p* < 0.001), respectively (Figure 1; Appendix A). In addition, multivariate Cox analysis also revealed that both OS-PII (HR: 1.407, 95% CI: 1.182–1.674, *p* < 0.001) and DFS-PII (HR: 1.162, 95% CI: 1.025–1.318, *p* = 0.019) were independent prognostic predictors for CRC patients in the validation cohort (Table 4 and Table 5).

### 2.5. Prognostic Value of Different Combinations of PIIs and TNM Staging

The OS and DFS rates of patients with stage II CRC were significantly higher than that of patients with stage III CRC in the training cohort (Appendix A). In addition, patients in the high OS-PII and high DFS-PII groups had lower OS and DFS rates in both stage II and stage III (Appendix A). Next, we assessed the association of OS-PII and DFS-PII with prognosis according to different TNM staging.

The stratification by a combination of PIIs and TNM staging divided patients into four risk groups (RG): RG1 (low PIIs and stage II), RG2 (high PIIs and stage II), RG3 (low PIIs and stage III), and RG4 (high PIIs and stage III). Kaplan–Meier curves showed that patients in the different RGs demonstrated significantly different survival (Figure 2A,B). Multivariate Cox models adjusting for clinicopathological factors demonstrated that, compared with patients in the RG1 group, the prognosis of patients in the RG2, RG3, and RG4 groups became worse and worse (Figure 2C,D, *p* for trend <0.001).

These findings were validated in an external validation cohort, in which patients were also divided into four RGs by a combination of PIIs and TNM staging (Appendix A). After being adjusted for significant clinicopathological factors, patients had worse prognosis as RGs increased (Appendix A, *p* for trend <0.001).

### 2.6. Prognostic Effects of OS-PII and DFS-PII in Different Subgroups

In the training cohort, the prognostic effects of OS-PII and DFS-PII among different subgroups defined by age, gender, tumor location, tumor diameter, CA19-9, and postoperative chemotherapy (no or yes) were not significantly different (Appendix A). In the validation cohort, the prognostic effects of OS-PII and DFS-PII were also consistent in age <60 or ≥60, male or female, colon or rectal cancer, tumor diameter <50 mm or ≥50 mm, CA19-9 <37 U/mL or ≥37 U/mL, and whether postoperative chemotherapy was taken or not (Appendix A).

### 2.7. Comparison of the Prognostic Accuracy of PIIs, TNM Staging, Their Combination, and Previously Reported Biomarkers

Time-dependent area under the curves (AUCs) associated with OS and DFS were generated to compare the sequential trends of PIIs, TNM staging, their combination, and biomarkers previously reported (NLR, PLR, LMR, and SII). The details of the AUC values of the above markers are listed in Appendix A. In the training cohort, both OS-PII and DFS-PII had a stable prognostic performance at various follow-up times, and their AUCs tended to be higher than the NLR, SII, PLR, and LMR throughout the observation period (Figure 3). The results of tests for comparing the time-dependent AUCs of PIIs with NLR, SII, PLR, and LMR showed that both OS-PII and DFS-PII had better accuracy in terms of prognosis prediction (Appendix A). However, in the validation cohort, time-dependent AUCs among NLR, PLR, LMR, SII, and PIIs did not show significant differences (Appendix A). Compared with TNM staging alone, the prognostic ability was better with a combination of PIIs and TNM staging. The results were also validated in the external cohort (Appendix A).

### 2.8. Development and Validation of Nomograms

The nomograms based on OS-PII (Figure 4A) and DFS-PII (Figure 5A) were generated for personalized survival prediction of CRC patients. The concordance index (C-index) for OS and DFS prediction were 0.718 (95% CI: 0.704–0.731) and 0.700 (95% CI: 0.684–0.716), respectively, in the training cohort, and similar results were observed when we used bootstrapping for internal validation (0.714 and 0.694). The C-index for OS and DFS prediction were 0.765 (95% CI: 0.745–0.785) and 0.698 (95% CI: 0.681–0.715), respectively, in the validation cohort, and were 0.759 and 0.693 in the internal validation. Compared with AJCC (American Joint Committee on Cancer) system, nomograms had higher C-index (Appendix A). The calibration curves for the postoperative 3-year OS (Figure 4B,C) and DFS (Figure 5B,C) showed high agreement between the prediction by nomograms and actual observations.

### 2.9. Decision Curve Analysis

Decision curve analyses for the prognostic models of the AJCC system and nomograms indicated that both two prognostic models showed a positive net benefit in predicting 5-year OS and DFS in the training and validation cohorts (Appendix A). Compared with the AJCC system, the nomograms have better clinical applicability because of their wider range of threshold probabilities and higher net benefit (Appendix A). The developed nomograms were worth using in terms of personalized survival prediction of patients with CRC.

## 3. Discussion

In this double-center, large sample retrospective cohort study, we systemically assessed the prognostic effects of preoperative peripheral blood platelet, neutrophil, lymphocyte, monocyte, eosinophil, and basophil counts in patients with CRC, and constructed a novel inflammatory index by integrating blood cell counts significantly associated with prognosis. Both OS-PII and DFS-PII were independent prognostic predictors for CRC patients and could separate patients into low-risk and high-risk groups. In the prognosis prediction of patients with CRC, both OS-PII and DFS-PII had a stable performance at different time points and could enhance the prognostic ability of TNM staging by combination. Furthermore, the nomograms including PIIs and clinicopathological characteristics could provide personalized OS and DFS prediction and help clinicians to identify high-risk populations. All these findings have been validated in an independent external cohort.

Inflammation, as an important hallmark of cancer [14], plays a critical role in all stages of tumorigenesis [11,12,15,26,27]. CRC is inflammation-driven cancer, as is known that inflammatory bowel diseases increase the risk of CRC [14,28]. Peripheral blood cell counts could reflect the extent of systemic inflammation and are readily available in clinical practice. Thus, inflammatory markers constructed based on blood cell counts, have been considered as potential prognostic predictors for CRC patients. Studies have mainly been focused on markers such as NLR, PLR, LMR, and SII, with the results showing that elevated NLR, PLR, and SII, and reduced LMR were associated with poor prognosis in CRC [19,20,21,22,29,30,31,32,33,34,35,36,37]. However, these markers were simply based on the ratio of two or three types of blood cell counts, which may not accurately provide information on the status of inflammation. In addition, these markers have not been validated in an external cohort. This context highlights that the development of a comprehensive peripheral blood marker is urgently needed for the identification of patients with CRC with high risk.

Neutrophils, as the first responders to inflammation, can migrate toward the tumor by CXC-chemokines [38,39] and promote the spread of cancer cells to distant organs [40]. Monocytes in the tumor microenvironment have the potential to differentiate into dendritic cells or tumor-associated macrophages [41,42]. Lymphocytes are important components for adaptive immunity and they suppress cancer cell proliferation by inducing apoptosis and inhibiting cancer cell migration and invasion [43]. The effects of platelets [44,45] and eosinophils [46,47] in cancer development and progression are controversial, whereas the effect of basophils on cancer is unknown. However, the relationship between the six types of peripheral blood cells and the prognosis of patients with CRC has not been systematically and clearly elucidated yet. We, for the first time, simultaneously investigated the potential relationship between these six types of blood cells and the prognosis of CRC. The results showed that platelets and eosinophils had a nonlinear relationship with both OS and DFS, which can explain the controversial roles of platelets and eosinophils in cancer; lymphocytes had a nonlinear relationship with OS, but were not associated with DFS, which also indirectly validated the results reported before [48]; and neutrophils and monocytes showed positive linear associations with OS and DFS, but basophils were not associated with the prognosis.

Based on the prognostic effects of different types of peripheral blood cell counts, we constructed the OS-PII and DFS-PII, respectively. The OS-PII and DFS-PII we presented here are innovative because they contained all the types of blood cells associated with prognosis and were integrated based on a weighted approach. Different from most of the published studies that assessed the prognosis of CRC using categorical variables of markers, we found that both continuous and binary variables of the OS-PII and DFS-PII are significant independent prognostic predictors (Table 2, Table 3, Table 4 and Table 5). Importantly, the prognostic value of OS-PII and DFS-PII were successfully validated in an independent cohort.

The CRC 5-year relative survival ranges from greater than 90% in patients with stage I to slightly greater than 10% in patients with stage IV [49]. Although TNM staging provides valuable prognostic information, the outcome of individual patients is not predicted accurately. This is a drawback for patients with stage II and III CRC in particular but also reminds us of the importance of developing well-performed markers. These significant biomarkers can help identify populations at high risk for recurrence or death in stage II and III patients. We developed the PIIs in a large sample, including patients with stage II and III CRC with sufficient follow-up. After stratification of patients by a combination of PIIs and TNM staging, we found that both OS-PII and DFS-PII had the ability to independently identify high-risk populations in the same TNM staging.

According to the results of time-dependent ROC analyses in the training cohort, the PIIs were superior to NLR, PLR, LMR, and SII in prediction. Unfortunately, this was not validated in the external independent cohort. Considering that the proportion of TNM staging in the two cohorts was inconsistent, time-dependent ROC analyses stratified by TNM staging in both training and validation cohorts were further performed. We found that the time-dependent AUCs showed little difference between stage II and stage III in the training cohort, but the difference became larger in the validation cohort (Appendix A). The differential proportion of TNM staging in the two cohorts may affect the prediction ability evaluation of PIIs. Collectively, the PIIs were not inferior to reported biomarkers and could be used as a marker for tumor progression and prognosis prediction in patients with CRC.

Nomograms, which can predict an individual’s probability of a clinical event by integrating diverse prognostic variables, are widely used in oncology [50]. Nomograms have been validated to compare favorably to the conventional TNM staging systems in many cancers [51,52]. Our study has built nomograms that assign predictions for survival based on both PIIs and significant clinicopathological factors. The nomograms performed well for predicting both OS (C-index: 0.718) and DFS (C-index: 0.700). The performance of the nomograms was verified in internal and external validations, which also implies the reliability of these nomograms. AJCC system is considered to be the benchmark for classifying patients with cancer and defining prognosis [53]. Compared with AJCC staging, the nomograms had higher C-index and net benefit, which implies the better clinical applicability of these nomograms.

To the best of our knowledge, this is the first attempt to construct a multinomial peripheral blood marker with all possible blood cell counts to identify high-risk populations with respect to recurrence or death in patients with CRC. Furthermore, this study has a large sample in both the training and external validation cohort. Our study also has several limitations. First, the PIIs were constructed and validated in two independent cohorts that included Chinese CRC patients. It would be better if multi-center validation can be carried out to verify whether the PIIs are universally applicable in other ethnicities. Second, our study found that the PIIs can independently and stably predict the prognosis of patients with stage II and stage III CRC. More studies should be conducted to evaluate whether the PIIs are also effective and feasible in patients with stage I and stage IV CRC. Third, our study was a retrospective cohort. Therefore it comes with a limitation that some data on clinicopathological characteristics are lacking, such as lymphovascular invasion, tumor budding, tumor-infiltrating lymphocyte, and microsatellite instability.

## 4. Materials and Methods

### 4.1. Study Population

A total of 11,127 primary stage II and III CRC patients confirmed by pathological diagnosis were enrolled in this study, including 4392 patients obtained from the Third Affiliated Hospital of Harbin Medical University, between January 2007 and December 2013 as the training cohort and 6735 patients obtained between January 2007 and December 2015 from the Fudan University Shanghai Cancer Center (FUSCC) as the external validation cohort. All these patients underwent radical resection surgery. A total of 1812 patients who met one or more of the following exclusion criteria were excluded (Figure 6): patients with age less than 18 years (*n* = 1); missing data on preoperative peripheral blood cell counts (*n* = 141); received neoadjuvant chemotherapy or other radiotherapy/chemotherapy before surgery (*n* = 883); and patients lost to follow-up within 3 months *(n* = 787). Finally, 4154 and 5161 CRC patients were included in the training and validation cohort, respectively. This study complied with the standards of the Helsinki Declaration. Throughout this article, the term “prognostic marker” is defined according to REMARK Guidelines [54].

### 4.2. Data Collection

Data of patients’ demographic and clinicopathological characteristics were obtained from retrospective medical records, including age, gender, BMI, history of hypertension and diabetes, tumor location, tumor diameter, pathological classification, differentiation degree, histologic classification, T and N stage, etc. Blood routine tests, which were based upon a single blood sample of each patient, were measured by an autoanalyzer (Sysmex XE-2100, Kobe, Japan). Data on peripheral blood cell counts including platelet, neutrophil, lymphocyte, monocyte, eosinophil, and basophil were extracted from the results of the first blood routine tests (limit to 30 days prior to surgery).

Patients were followed up regularly in accordance with NCCN guidelines. The last time of follow-up for the training and external validation cohorts was 22 January 2019, and 18 July 2020, respectively. The survival information was obtained from the hospital medical records follow-up platform or contacts with patients by phone or email. OS was defined as the period from surgery to death from any cause, or the last contact. DFS was defined as the period from surgery to local recurrence, distant metastasis, a new primary tumor of CRC, or death, whichever comes first.

### 4.3. Construction of the PII

Prognostic factors of interest for constructing the PII were platelet, neutrophil, lymphocyte, monocyte, eosinophil, and basophil counts. The PII was constructed in the training cohort based on the following steps. First, RCS regression was performed to determine whether there was a nonlinear relationship between six types of blood cell counts and survival (OS and DFS). Factors that had significant nonlinear relationships with OS or DFS (*p* < 0.05) were converted into binary variables using X-tile 3.6.1 software [55] (Yale University, New Haven, CT, USA), while the others as continuous variables, without conversion. Second, univariate Cox proportional hazards models were implemented to investigate the association between the six types of blood cell counts and OS or DFS. Significant prognostic factors (*p* < 0.10) in the univariate analyses were then entered into multivariate Cox proportional hazards models. Finally, OS-PII and DFS-PII were constructed using blood cell counts associated with OS and DFS, with weights given by the corresponding coefficients from the multivariate Cox model.

### 4.4. Statistical Analyses

The missing data of included variables were filled in by using the multiple imputation method [56]. Student’s *t*-tests for normally distributed continuous variables, χ^2^ tests for categorical variables, and Mann–Whitney U tests for non-normally distributed continuous variables were used to evaluate the differences between training and validation cohorts. X-tile 3.6.1 software [55] was used to determine the optimal cut-off values for the PIIs in the training cohort, which were also used in the external validation cohort. The χ^2^ tests were performed to assess the association between the PIIs and clinicopathological characteristics. The 1-, 3-, 5- and 10-year OS and DFS were calculated using the Kaplan–Meier method, and the difference between two PII levels was compared using log-rank tests. The prognostic value of clinicopathological characteristics and PIIs were estimated using univariate and multivariate Cox analyses. The results of Cox analyses were presented as HR and 95% CI. Additionally, subgroup analyses were also conducted, stratified by age (<60 years old; ≥60 years old), gender (male; female), tumor location (colon; rectum), tumor diameter (<50 mm; ≥50 mm), CA19-9 (<37 U/mL; ≥37 U/mL), and postoperative chemotherapy (no or yes).

The nomograms for possible prognostic factors associated with OS and DFS were developed to predict the probability of 1-, 3- and 5-year survival recurrence/metastasis for CRC patients. Multivariate Cox analyses determined the effects of prognostic factors on a nomogram, and only the factors with a *p*-value < 0.05 were finally incorporated into the nomogram. The prediction accuracy of the nomograms was evaluated by the C-index [50,57]. The value of the C-index ranged from 1.0 (perfect concordance) to 0.0 (perfect discordance), and a value of 0.5 is equivalent to a random chance. Nomograms map the predicted probabilities into points on a scale from 0 to 100, and can be interpreted by accumulating the points correspond to the predicted probability, which is indicated at the top of the scale [57]. Bootstrapping techniques were used for internal validation of the prognostic models, and the calibration of nomograms was assessed graphically by plotting the actual probabilities versus the nomogram-predicted probabilities [50,57]. Decision curve analyses were also performed for the prognostic models of nomograms and AJCC system. The net benefits of nomograms and AJCC system were compared to evaluate the clinical applicability of these two models. Time-dependent ROC analyses were performed, and the estimated AUCs were calculated to compare the prognostic abilities of the PIIs, TNM staging, their combination, and biomarkers previously reported (NLR, PLR, LMR, and SII) [58,59].

All statistical analyses were performed with SPSS 24.0 (SPSS Inc., Chicago, IL, USA) and R 3.6.2 software (Institute for Statistics and Mathematics, Vienna, Austria). Two sided *p* < 0.05 was considered statistically significant.

## 5. Conclusions

We constructed a novel PII by integrating all the preoperative peripheral blood cell counts associated with prognosis and systematically analyzed the role of PIIs in the prognosis of CRC. Our study demonstrates that both OS-PII and DFS-PII are independent factors for predicting the prognosis of CRC and could enhance the prognostic ability of TNM staging by combination. The nomograms based on OS and DFS can be recommended for the personalized survival prediction of patients with CRC.

## Figures and Tables

**Figure 1 cancers-13-00003-f001:**
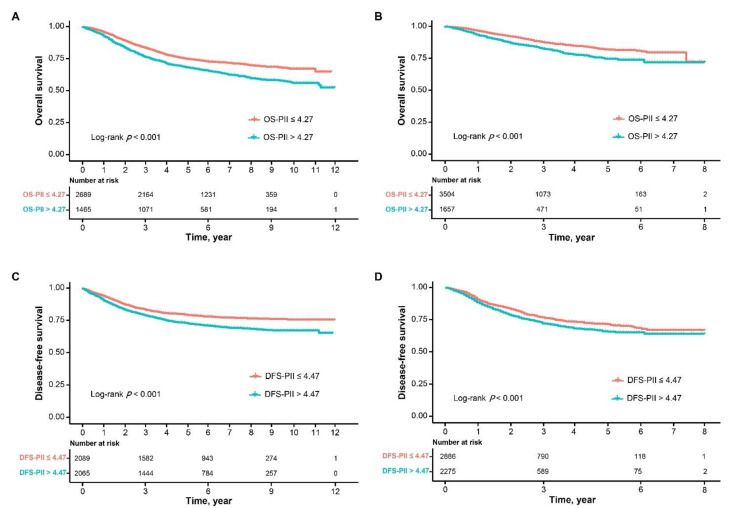
Kaplan–Meier curves of overall survival and disease-free survival for OS-PII and DFS-PII in patients with CRC. Kaplan–Meier curves of overall survival for OS-PII in the training cohort (**A**), and for OS-PII in the validation cohort (**B**), and Kaplan–Meier curves of disease-free survival for DFS-PII in the training cohort (**C**), and for DFS-PII in the validation cohort (**D**).

**Figure 2 cancers-13-00003-f002:**
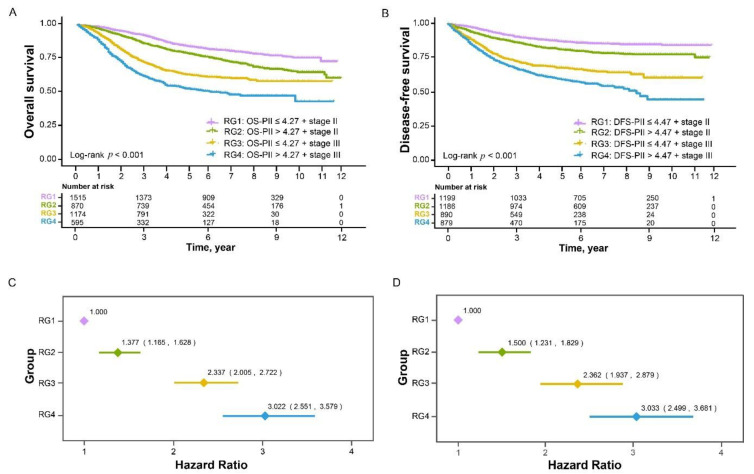
Risk stratification combining PIIs and TNM staging in relation to overall survival and disease-free survival of CRC in the training cohort. Kaplan–Meier curves of four risk groups for overall survival (**A**) and disease-free survival (**B**). Multivariate Cox analyses of the four risk groups for overall survival (**C**) and disease-free survival (**D**), adjusting for the significant clinicopathological factors in relation to overall survival (Table 2) and disease-free survival (Table 3).

**Figure 3 cancers-13-00003-f003:**
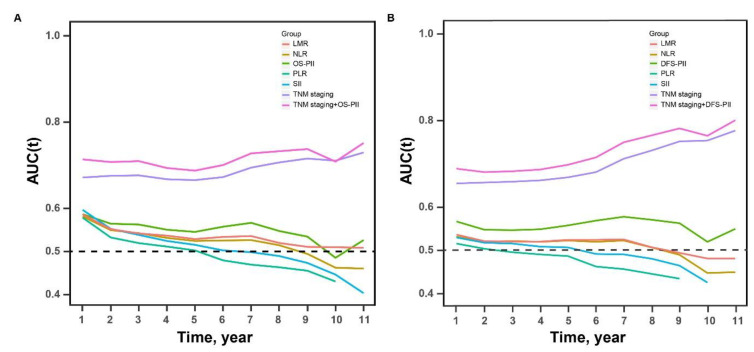
The time-dependent AUCs of PIIs, TNM staging, a combination of PIIs and TNM staging, NLR, PLR, LMR, and SII in the training cohort. Time-dependent AUCs presented the sequential trends of PIIs, TNM staging, a model of PIIs and TNM staging, NLR, PLR, LMR, and SII for overall survival prediction (**A**) and disease-free survival prediction (**B**). The horizontal axis represents the years after radical resection, and the vertical axis represents the estimated area under the ROC curves for survival at the time of interest.

**Figure 4 cancers-13-00003-f004:**
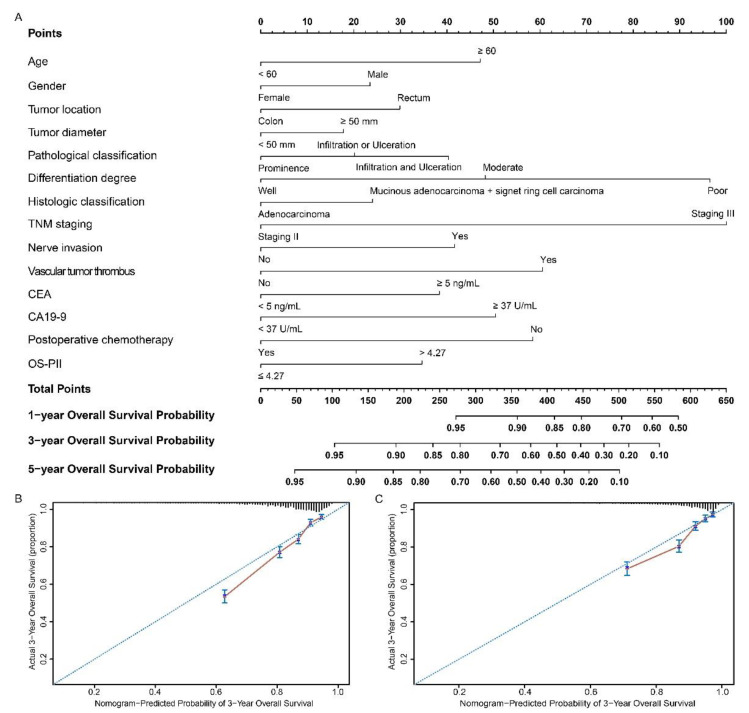
Nomograms to predict overall survival in patients with CRC. Nomograms were performed by using clinicopathological characteristics and OS-PII to predict overall survival (**A**) and calibration curves of the nomogram to predict overall survival at 3 years in the training cohort (**B**) and the validation cohort (**C**). Nomograms map the predicted probabilities onto points on a scale from 0 to 100 and can be interpreted by accumulating the points that correspond to the predicted probability, which is indicated at the top of scale. The total points were converted to predict 1-, 3- and 5-year probabilities of death and recurrence or metastasis for patients with CRC.

**Figure 5 cancers-13-00003-f005:**
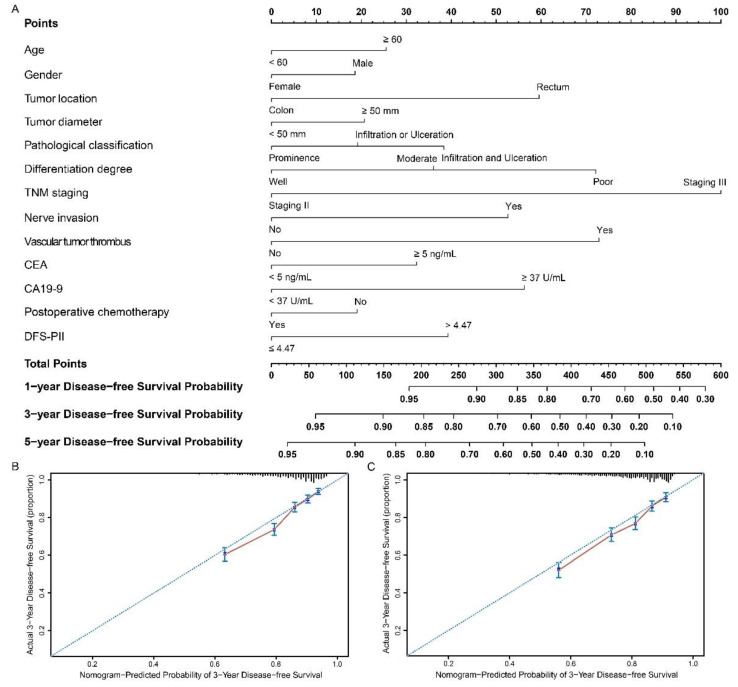
Nomograms to predict disease-free survival in patients with CRC. Nomograms were performed by using clinicopathological characteristics and DFS-PII to predict disease-free survival (**A**) and calibration curves of the nomogram to predict disease-free survival at 3 years in the training cohort (**B**) and the validation cohort (**C**). Nomograms map the predicted probabilities onto points on a scale from 0 to 100 and can be interpreted by accumulating the points that correspond to the predicted probability, which is indicated at the top of scale. The total points were converted to predict 1-, 3- and 5-year probabilities of death and recurrence or metastasis for patients with CRC.

**Figure 6 cancers-13-00003-f006:**
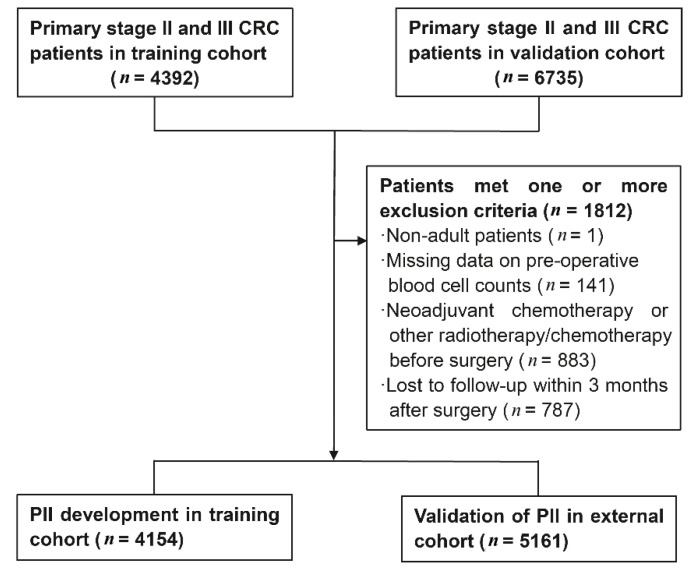
Flowchart of study patient selection.

**Table 1 cancers-13-00003-t001:** Patients’ baseline characteristics in the training and validation cohorts.

Demographic or Characteristic	Training Cohort (*N* = 4154)	Validation Cohort (*N* = 5161)	*p* Value
Age (year) ^a^	59.3 ± 11.65	58.5 ± 11.95	0.850
<60	2103 (50.6)	2645 (51.2)	0.549
≥60	2051 (49.4)	2516 (48.8)	
Gender ^b^			0.804
Male	2454 (59.1)	3062 (59.3)	
Female	1700 (40.9)	2099 (40.7)	
BMI (kg/m^2^) ^b^			-
<24	1828 (44.0)	-	
≥24	1383 (33.3)	-	
Hypertension ^b^			-
No	3554 (85.6)	-	
Yes	600 (14.4)	-	
Diabetes mellitus ^b^			-
No	3842 (92.5)	-	
Yes	312 (7.5)	-	
Tumor location ^b^			0.005
Right colon	899 (21.6)	1220 (23.7)	
Transverse colon	85 (2.0)	110 (2.1)	
Left colon	256 (6.2)	362 (7.0)	
Sigmoid colon	721 (17.4)	773 (15.0)	
Rectum	2193 (52.8)	2696 (52.2)	
Tumor diameter ^b^			<0.001
<50 mm	1640 (39.5)	3446 (66.8)	
≥50 mm	2359 (56.8)	1710 (33.1)	
Pathological classification ^b^			<0.001
Prominence	2740 (66.0)	1277 (24.7)	
Infiltration	268 (6.4)	238 (4.6)	
Ulceration	160 (3.9)	3436 (66.6)	
Infiltration and ulceration	986 (23.7)	210 (4.1)	
Differentiation degree ^b^			<0.001
Well	331 (8.0)	48 (0.9)	
Moderate	3225 (77.6)	3755 (72.8)	
Poor	598 (14.4)	1358 (26.3)	
Histologic classification ^b^			<0.001
Adenocarcinoma	3136 (75.5)	4342 (84.1)	
Mucinous adenocarcinoma + signet ring cell carcinoma	1018 (24.5)	819 (15.9)	
TNM staging ^b^			<0.001
II	2385 (57.4)	2086 (40.4)	
III	1769 (42.6)	3075 (59.6)	
AJCC staging II ^b^			<0.001
IIA	974 (40.8)	1175 (56.3)	
IIB	180 (7.6)	882 (42.3)	
IIC	1231 (51.6)	29 (1.4)	
AJCC staging III ^b^			<0.001
IIIA	122 (6.9)	271 (8.8)	
IIIB	752 (42.5)	1727 (56.2)	
IIIC	895 (50.6)	1077 (35.0)	
Tumor invasion ^b^			<0.001
T1–T3	1847 (44.5)	2588 (50.1)	
T4	2307 (55.5)	2573 (49.9)	
Lymph nodes involved ^b^			<0.001
N0	2385 (57.4)	2086 (40.4)	
N1–N2	1769 (42.6)	3075 (59.6)	
Cancer nodules ^b^			<0.001
No	3863 (93.0)	4249 (82.3)	
Yes	291 (7.0)	912 (17.7)	
Nerve invasion ^b^			<0.001
No	3836 (92.3)	3898 (75.5)	
Yes	318 (7.7)	1263 (24.5)	
Vascular tumor thrombus ^b^			<0.001
No	4009 (96.5)	3622 (70.2)	
Yes	145 (3.5)	1539 (29.8)	
CEA ^b^			0.971
<5 ng/mL	2203 (53.0)	2832 (54.9)	
≥5 ng/mL	1626 (39.1)	2087 (40.4)	
CA19-9 ^b^			<0.001
<37 U/mL	3026 (72.8)	3932 (76.2)	
≥37 U/mL	621 (14.9)	991 (19.2)	
Postoperative chemotherapy ^b^			<0.001
No	2413 (58.1)	940 (18.2)	
Yes	1741 (41.9)	4221 (81.8)	
Postoperative radiotherapy ^b^			<0.001
No	3969 (95.5)	4752 (92.1)	
Yes	185 (4.5)	409 (7.9)	
Platelet counts (10^9^/L) ^c^	247 (204–305)	232 (189–283)	<0.001
Neutrophil counts (10^9^/L) ^c^	3.77 (2.95–4.83)	3.50 (2.80–4.50)	<0.001
Lymphocyte counts (10^9^/L) ^c^	1.89 (1.50–2.34)	1.70 (1.30–2.10)	<0.001
Monocyte counts (10^9^/L) ^c^	0.43 (0.33–0.54)	0.40 (0.30–0.50)	<0.001
Eosinophil counts (10^9^/L) ^c^	0.12 (0.06–0.20)	0.13 (0.08–0.22)	<0.001
Basophil counts (10^9^/L) ^c^	0.04 (0.02–0.06)	0.02 (0.01–0.04)	<0.001

Abbreviations: BMI, body mass index; AJCC, American Joint Committee on Cancer; Data are presented as ^a^ mean (standard deviation), ^b^ n (%) or ^c^ median (interquartile ranges). The validation cohort did not contain BMI and history of hypertension and diabetes. The number of missing values for BMI, tumor diameter, CEA, and CA19-9 were 943, 155, 325 and 507, respectively, in the training cohort. The number of missing values for tumor diameter, CEA, and CA19-9 were 5, 242 and 238, respectively, in the validation cohort.

**Table 2 cancers-13-00003-t002:** Association between predictive factors and overall survival of CRC in the training cohort.

Demographic or Characteristic	Univariate Analysis		Multivariate Analysis	
	HR (95% CI)	*p* Value	HR (95% CI)	*p* Value
Age		<0.001		<0.001
<60	1.000		1.000	
≥60	1.654 (1.479–1.851)		1.521 (1.354–1.708)	
Gender		0.013		<0.001
Male	1.000		1.000	
Female	0.866 (0.773–0.970)		0.802 (0.715–0.899)	
Tumor location		<0.001		0.001
Colon	1.000		1.000	
Rectum	1.297 (1.160–1.449)		1.224 (1.089–1.377)	
Tumor diameter		<0.001		0.009
<50 mm	1.000		1.000	
≥50 mm	1.248 (1.113–1.400)		1.171 (1.040–1.319)	
Pathological classification				
Prominence	1.000		1.000	
Infiltration or Ulceration	1.544 (1.298–1.836)	<0.001	1.357 (1.138–1.619)	0.001
Infiltration and Ulceration	1.578 (1.394–1.786)	<0.001	1.394 (1.228–1.581)	<0.001
Differentiation degree				
Well	1.000		1.000	
Moderate	1.608 (1.243–2.079)	<0.001	1.480 (1.142–1.916)	0.003
Poor	2.812 (2.132–3.708)	<0.001	2.269 (1.715–3.003)	<0.001
Histologic classification		0.011		0.002
Adenocarcinoma	1.000		1.000	
Mucinous adenocarcinoma or signet ring cell carcinoma	1.176 (1.039–1.332)		1.219 (1.073–1.385)	
TNM staging		<0.001		<0.001
II	1.000		1.000	
III	2.447 (2.187–2.738)		2.248 (1.995–2.534)	
Nerve invasion		<0.001		0.001
No	1.000		1.000	
Yes	1.774 (1.480–2.126)		1.387 (1.146–1.679)	
Vascular tumor thrombus		<0.001		<0.001
No	1.000		1.000	
Yes	2.435 (1.922–3.087)		1.669 (1.302–2.139)	
CEA		<0.001		<0.001
<5 ng/mL	1.000		1.000	
≥5 ng/mL	1.709 (1.521–1.921)		1.373 (1.217–1.550)	
CA19-9		<0.001		<0.001
<37 U/mL	1.000		1.000	
≥37 U/mL	2.012 (1.761–2.298)		1.525 (1.322–1.760)	
Postoperative chemotherapy		<0.001		<0.001
No	1.000		1.000	
Yes	0.669 (0.596–0.751)		0.578 (0.511–0.654)	
Postoperative radiotherapy		<0.001		<0.001
No	1.000		1.000	
Yes	1.854 (1.496–2.298)		1.824 (1.458–2.282)	
OS-PII (Continuous)	1.105 (1.072–1.139)	<0.001	1.087 (1.052–1.122)	<0.001
OS-PII (Binary)		<0.001		<0.001
≤4.27	1.000		1.000	
>4.27	1.400 (1.253–1.565)		1.330 (1.189–1.489)	

Abbreviations: HR, hazard ratio; CI, confidence interval.

**Table 3 cancers-13-00003-t003:** Association between predictive factors and disease-free survival of CRC in the training cohort.

Demographic or Characteristic	Univariate Analysis		Multivariate Analysis	
	HR (95% CI)	*p* Value	HR (95% CI)	*p* Value
Age (year)		0.010		<0.001
<60	1.000		1.000	
≥60	1.176 (1.040–1.331)		1.261 (1.109–1.433)	
Gender		0.048		0.014
Male	1.000		1.000	
Female	0.880 (0.776–0.999)		0.851 (0.749–0.968)	
Tumor location		<0.001		<0.001
Colon	1.000		1.000	
Rectum	1.633 (1.437–1.856)		1.504 (1.314–1.721)	
Tumor diameter		0.004		0.008
<50 mm	1.000		1.000	
≥50 mm	1.211 (1.063–1.379)		1.195 (1.047–1.362)	
Pathological classification				
Prominence	1.000		1.000	
Infiltration or Ulceration	1.559 (1.288–1.888)	<0.001	1.395 (1.148–1.695)	0.001
Infiltration and Ulceration	1.498 (1.302–1.724)	<0.001	1.316 (1.141–1.518)	<0.001
Differentiation degree				
Well	1.000		1.000	
Moderate	1.336 (1.029–1.734)	0.030	1.257 (0.966–1.637)	0.088
Poor	2.161 (1.620–2.882)	<0.001	1.708 (1.275–2.287)	<0.001
Histologic classification		0.164		-
Adenocarcinoma	1.000		-	
Mucinous adenocarcinoma or signet ring cell carcinoma	1.105 (0.960–1.273)		-	
TNM staging		<0.001		<0.001
II	1.000		1.000	
III	2.720 (2.396–3.088)		2.148 (1.878–2.457)	
Nerve invasion		<0.001		<0.001
No	1.000		1.000	
Yes	2.084 (1.728–2.513)		1.479 (1.212–1.805)	
Vascular tumor thrombus		<0.001		<0.001
No	1.000		1.000	
Yes	2.810 (2.203–3.584)		1.758 (1.361–2.273)	
CEA		<0.001		0.001
<5 ng/mL	1.000		1.000	
≥5 ng/mL	1.530 (1.347–1.738)		1.259 (1.100–1.442)	
CA19-9		<0.001		<0.001
<37 U/mL	1.000		1.000	
≥37 U/mL	1.914 (1.631–2.245)		1.594 (1.353–1.878)	
Postoperative chemotherapy		<0.001		0.277
No	1.000		1.000	
Yes	1.272 (1.125–1.439)		1.076 (0.943–1.229)	
Postoperative radiotherapy		<0.001		<0.001
No	1.000		1.000	
Yes	3.212 (2.623–3.932)		2.281 (1.839–2.828)	
DFS-PII (Continuous)	1.105 (1.069–1.143)	<0.001	1.089 (1.053–1.128)	<0.001
DFS-PII (Binary)		<0.001		<0.001
≤4.47	1.000		1.000	
>4.47	1.395 (1.233–1.580)		1.366 (1.206–1.548)	

Abbreviations: HR, hazard ratio; CI, confidence interval.

**Table 4 cancers-13-00003-t004:** Association between predictive factors and overall survival of CRC in the validation cohort.

Demographic or Characteristic	Univariate Analysis		Multivariate Analysis	
	HR (95% CI)	*p* Value	HR (95% CI)	*p* Value
Age		<0.001		<0.001
<60	1.000		1.000	
≥60	1.418 (1.197–1.679)		1.468 (1.235–1.745)	
Gender		0.508		0.604
Male	1.000		1.000	
Female	1.059 (0.893–1.256)		1.047 (0.881–1.244)	
Tumor location		0.397		0.653
Colon	1.000		1.000	
Rectum	0.930 (0.786–1.100)		0.958 (0.793–1.156)	
Tumor diameter		0.603		0.288
<50 mm	1.000		1.000	
≥50 mm	1.048 (0.878–1.252)		1.106 (0.919–1.331)	
Pathological classification				
Prominence	1.000		1.000	
Infiltration or Ulceration	1.488 (1.189–1.861)	0.001	1.223 (0.972–1.539)	0.086
Infiltration and Ulceration	1.616 (1.081–2.415)	0.019	1.424 (0.948–2.139)	0.088
Differentiation degree				
Well	1.000		1.000	
Moderate	2.289 (0.570–9.196)	0.243	1.350 (0.334–5.452)	0.673
Poor	4.904 (1.218–19.745)	0.025	2.096 (0.516–8.513)	0.301
Histologic classification		0.001		0.634
Adenocarcinoma	1.000		1.000	
Mucinous adenocarcinoma or signet ring cell carcinoma	1.413 (1.145–1.743)		1.059 (0.838–1.338)	
TNM staging		<0.001		<0.001
II	1.000		1.000	
III	2.985 (2.414–3.691)		2.083 (1.648–2.634)	
Nerve invasion		<0.001		0.010
No	1.000		1.000	
Yes	1.950 (1.635–2.325)		1.284 (1.062–1.551)	
Vascular tumor thrombus		<0.001		<0.001
No	1.000		1.000	
Yes	2.774 (2.342–3.285)		1.721 (1.427–2.076)	
CEA		<0.001		<0.001
<5 ng/mL	1.000		1.000	
≥5 ng/mL	2.375 (1.994–2.830)		1.661 (1.376–2.006)	
CA19-9		<0.001		<0.001
<37 U/mL	1.000		1.000	
≥37 U/mL	3.073 (2.583–3.654)		1.917 (1.581–2.323)	
Postoperative chemotherapy		0.014		<0.001
No	1.000		1.000	
Yes	0.774 (0.631–0.949)		0.618 (0.499–0.766)	
Postoperative radiotherapy		0.016		0.089
No	1.000		1.000	
Yes	1.329 (1.055–1.675)		1.257 (0.965–1.637)	
OS-PII (Continuous)	1.164 (1.107–1.224)	<0.001	1.133 (1.076–1.194)	<0.001
OS-PII (Binary)		<0.001		<0.001
≤4.27	1.000		1.000	
>4.27	1.561 (1.316–1.852)		1.407 (1.182–1.674)	

Abbreviations: HR, hazard ratio; CI, confidence interval.

**Table 5 cancers-13-00003-t005:** Association between predictive factors and disease-free survival of CRC in the validation cohort.

Demographic or Characteristic	Univariate Analysis		Multivariate Analysis	
	HR (95% CI)	*p* Value	HR (95% CI)	*p* Value
Age (year)		0.291		0.108
<60	1.000		1.000	
≥60	1.070 (0.944–1.212)		1.110 (0.978–1.261)	
Gender		0.623		0.328
Male	1.000		1.000	
Female	0.969 (0.853–1.100)		0.938 (0.825–1.067)	
Tumor location		0.946		0.891
Colon	1.000		1.000	
Rectum	1.004 (0.888–1.138)		0.990 (0.863–1.136)	
Tumor diameter		0.266		0.951
<50 mm	1.000		1.000	
≥50 mm	0.927 (0.810–1.060)		0.996 (0.866–1.144)	
Pathological classification				
Prominence	1.000		1.000	
Infiltration or Ulceration	1.420 (1.208–1.668)	<0.001	1.193 (1.013–1.405)	0.035
Infiltration and Ulceration	1.414 (1.040–1.923)	0.027	1.232 (0.904–1.680)	0.187
Differentiation degree				
Well	1.000		1.000	
Moderate	1.796 (0.744–4.332)	0.192	1.149 (0.475–2.780)	0.758
Poor	2.965 (1.225–7.178)	0.016	1.435 (0.590–3.491)	0.426
Histologic classification		0.056		-
Adenocarcinoma	1.000		-	
Mucinous adenocarcinoma or signet ring cell carcinoma	1.174 (0.996–1.384)		-	
TNM staging		<0.001		<0.001
II	1.000		1.000	
III	2.448 (2.112–2.837)		1.684 (1.429–1.984)	
Nerve invasion		<0.001		<0.001
No	1.000		1.000	
Yes	1.999 (1.755–2.277)		1.389 (1.207–1.597)	
Vascular tumor thrombus		<0.001		<0.001
No	1.000		1.000	
Yes	2.249 (1.983–2.551)		1.528 (1.327–1.758)	
CEA		<0.001		<0.001
<5 ng/mL	1.000		1.000	
≥5 ng/mL	2.004 (1.763–2.279)		1.589 (1.384–1.824)	
CA19-9		<0.001		<0.001
<37 U/mL	1.000		1.000	
≥37 U/mL	2.306 (2.015–2.639)		1.584 (1.364–1.841)	
Postoperative chemotherapy		0.004		0.317
No	1.000		1.000	
Yes	1.291 (1.087–1.534)		1.096 (0.916–1.312)	
Postoperative radiotherapy		0.001		0.257
No	1.000		1.000	
Yes	1.353 (1.133–1.615)		1.121 (0.920–1.366)	
DFS-PII (Continuous)	1.054 (1.036–1.073)	0.003	1.037 (1.002–1.075)	0.040
DFS-PII (Binary)		0.001		0.019
≤4.47	1.000		1.000	
>4.47	1.248 (1.101–1.414)		1.162 (1.025–1.318)	

Abbreviations: HR, hazard ratio; CI, confidence interval.

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
