# Peer review of "Prognostic Inflammatory Index Based on Preoperative Peripheral Blood for Predicting the Prognosis of Colorectal Cancer Patients"

_cancers, 2020, doi:10.3390/cancers13010003_

Round 1

Reviewer 1 Report

In this paper the authors report the development of prognostic parameters for colorectal cancer based upon preoperative blood cell counts including lymphocytes, monocytes, neutrophils, eosinophils, and platelets. The study was conducted on existing patient cohorts using available data only. The retrospective approach comes with some limitations as some relevant data were not available (see later). The authors developed separate prognostic indices for overall survival (OS) and disease-free survival (DFS) as in the latter lymphocyte counts were not included in view of the lack of prognostic correlation between lymphocyte counts and DFS. Blood cell count based prognostic indices were validated in an independent patient cohort. Patients could be separated into prognostic groups based on a combination of TNM stage and PII. Finally a nomogram was developed allowing the inclusion of these new prognostic parameters with various other prognostically relevant data.

I have the following questions:

1.the retrospective character of the study comes with as important limitation that some important data are lacking. An important one is the exact timing of the blood sample. This is nowhere mentioned. Was this at a standard preoperative time frame? Was this based upon a single blood sample? Any data regarding evolution of blood cell counts in time?

  1. relevant pathological parameters which are missing are lymphovascular invasion and tumor budding. These have become standard parameters allowing with other parameters to separate Stage 2 into high and low risk subcategories. It would have been relevant to have included these notably in combining PII with TNM stage to get to the four G groups.
  2. information as to MSI status is missing. This is quite relevant when it comes to an inflammatory response as MSI CRC have a high TIL count and a high immunoscore and MSI status might have been found an important covariable.
  3. information as to tumor tissue TIL counts or immunoscore are lacking. We now have a phenomenological exercise in terms of development of inflammation associated blood cell count based prognostic parameters, without any information as to what might be going on at the tissue level. The combination of the two might have provided information as to the mechanisms involved.
  4. the latter is exemplified by the finding that lymphocyte counts did not correlate with DFS. This is in stark contrast with the observation that TIL counts or immunoscore have strong correlation with prognosis including DFS.

Author Response

Response to Reviewer 1 Comments

Point 1: The retrospective character of the study comes with as important limitation that some important data are lacking. An important one is the exact timing of the blood sample. This is nowhere mentioned. Was this at a standard preoperative time frame? Was this based upon a single blood sample? Any data regarding evolution of blood cell counts in time?

Response 1: Thank you for your insightful comments and questions. We feel very sorry that we ignored to describe the exact timing of the blood sample. In our study, data of peripheral blood cell counts including platelet, neutrophil, lymphocyte, monocyte, eosinophil, and basophil were extracted from the results of the first blood routine tests that were based upon a single blood sample of each patient and the timing of the blood sample was limited to 30 days prior to surgery. If the interval between the timing of the blood sample and the timing of the operation is more than 30 days, the patients were excluded because of missing data on preoperative blood cell count. And in our study, most of the blood routine tests (99%) were performed within 15 days before surgery. We agree with the reviewer and have supplemented the information about the timing of the blood sample “Blood routine tests that were based upon a single blood sample of each patient, were measured by an autoanalyzer (Sysmex XE-2100, Kobe, Japan). Data on peripheral blood cell counts including platelet, neutrophil, lymphocyte, monocyte, eosinophil, and basophil were extracted from the results of the first blood routine tests (limit to 30 days prior to surgery)” in the Materials and Methods section (see line 374-378 on page 17 in the manuscript). We have already collected the data on both preoperative and postoperative blood routine tests. However, considering that the patients may take antibiotics after the preoperative first blood routine test and in the postoperative treatment, which may affect the blood cell counts and systemic inflammatory status. Therefore, we finally chose the data on the first blood routine test before surgery.

Point 2: Relevant pathological parameters which are missing are lymphovascular invasion and tumor budding. These have become standard parameters allowing with other parameters to separate Stage 2 into high and low risk subcategories. It would have been relevant to have included these notably in combining PII with TNM stage to get to the four G groups.

Response 2: Thank you for your valuable comment and suggestion. As the reviewer said, the retrospective character of the study comes with a limitation that some data are lacking. In our study, the data on clinicopathological characteristics were extracted from postoperative pathological reports. Because of the earlier time this study started, the postoperative pathological reports did not contain data on lymphovascular invasion and tumor budding. However, the Cox analysis performed in our study included vascular tumor thrombus, which could also reflect the prognostic effect of lymphovascular invasion. The results of multivariate Cox analyses showed that vascular tumor thrombus was statistically associated with both the OS and DFS in the training and validation cohorts (see Table 2-5 in the manuscript). Therefore, vascular tumor thrombus was entered into multivariate Cox analyses to adjust the prognostic effect of PIIs. We also added the retrospective limitations of our study “Third, our study was a retrospective cohort. Therefore it comes with a limitation that some data on clinicopathological characteristics are lacking, such as lymphovascular invasion, tumor budding, tumor-infiltrating lymphocyte, and microsatellite instability” in the discussion section (see line 350-353 on page 16 in the manuscript). Thanks for your comments that could make our paper more accurate.

Point 3: Information as to MSI status is missing. This is quite relevant when it comes to an inflammatory response as MSI CRC have a high TIL count and a high immunoscore and MSI status might have been found an important covariable.

Response 3: Thank you for your comment. Actually, our retrospective cohort study had a limitation that is some clinicopathological data are lacking. In the clinical practice of cancer treatment, MSI status is not a routine examination. Therefore, we cannot extract the data on MSI status of each patient from retrospective medical records. Our laboratory tested the MSI status of a small part of CRC patients previously, but these data was not enough to support the survival analyses of this study. We feel sorry about this, and we added the retrospective limitations of our study “Third, our study was a retrospective cohort. Therefore it comes with a limitation that some data on clinicopathological characteristics are lacking, such as lymphovascular invasion, tumor budding, tumor-infiltrating lymphocyte, and microsatellite instability” in the discussion section (see line 350-353 on page 16 in the manuscript).

Point 4: Information as to tumor tissue TIL counts or immunoscore are lacking. We now have a phenomenological exercise in terms of development of inflammation associated blood cell count based prognostic parameters, without any information as to what might be going on at the tissue level. The combination of the two might have provided information as to the mechanisms involved.

Response 4: Thank you for your considerable and valuable comment and suggestion. You are right. Our study is a phenomenological exercise in terms of the construction of prognostic inflammatory index (PII) by integrating the blood cell counts associated with prognosis. According to the results of our study, this novel PII could predict the survival of CRC patients independently and identify high-risk populations. However, the mechanism of the prognostic value of the PIIs in cancer remains unclear. Your viewpoint that is “combined tumor-infiltrating lymphocyte with biomarkers based on peripheral blood cell counts might provide and explain the mechanism of the prognostic value of the PIIs” is very logical and valuable. Because of the earlier time this cohort started, the tumor tissue TIL counts were not tested at that time. We could not obtain the information as to tumor tissue TIL counts or immune-score retrospectively. In further studies, we will collect the information at the tissue level, in order to explore the mechanism of the prognostic value of the inflammatory biomarkers based on peripheral blood cell counts.

Point 5: The latter is exemplified by the finding that lymphocyte counts did not correlate with DFS. This is in stark contrast with the observation that TIL counts or immunoscore have strong correlation with prognosis including DFS.

Response 5: Thank you for your insightful comment. In our study, we found that lymphocyte counts were associated with OS but did not correlate with DFS. This seems to be in stark contrast with the observation that tumor tissue TIL counts or immune-score have a strong correlation with the prognosis of CRC including DFS. Actually, T lymphocytes, especially CD8+ cytotoxic T cells, are considered to be the main anti-tumor immune effector cells. More TIL in tumor tissue is usually associated with a better prognosis [1]. These TILs are derived from peripheral blood specific lymphocyte subset populations, such as CD3+, CD4+and CD8+ T cells, etc. In addition, McMillan et al found that reduction of peripheral blood CD4+ T-lymphocytes occurs before the detectable recurrence of colorectal cancer [2], which implied the change of lymphocyte subset populations may reflect the progression of colorectal cancer. Absolute lymphocyte counts represents only a small portion of the whole lymphocyte pool and do not provide information regarding the proportion of different lymphocyte subset populations. Therefore, even if peripheral blood specific lymphocyte subset populations may be associated with DFS in colorectal cancer, this prognostic effect may not be reflected through the absolute lymphocyte counts, which could explain the above contrast. Therefore, peripheral blood specific lymphocyte subset populations may be more suitable factor for DFS prediction in colorectal cancer. In addition to our study, previous studies also have found that lymphocyte counts were associated with OS but did not correlate with DFS [3, 4].

  1. von Kleist, S.; Berling, J.; Bohle, W.; Wittekind, C. Immunohistological analysis of lymphocyte subpopulations infiltrating breast carcinomas and benign lesions. Int J Cancer 1987, 40, 18-23, doi:10.1002/ijc.2910400105.
  2. McMillan, D.C.; Fyffe, G.D.; Wotherspoon, H.A.; Cooke, T.G.; McArdle, C.S. Prospective study of circulating T-lymphocyte subpopulations and disease progression in colorectal cancer. Dis Colon Rectum 1997, 40, 1068-1071, doi:10.1007/bf02050931.
  3. Kozak, M.M.; von Eyben, R.; Pai, J.S.; Anderson, E.M.; Welton, M.L.; Shelton, A.A.; Kin, C.; Koong, A.C.; Chang, D.T. The Prognostic Significance of Pretreatment Hematologic Parameters in Patients Undergoing Resection for Colorectal Cancer. Am J Clin Oncol 2017, 40, 405-412, doi:10.1097/coc.0000000000000183.
  4. Oh, S.Y.; Heo, J.; Noh, O.K.; Chun, M.; Cho, O.; Oh, Y.T. Absolute Lymphocyte Count in Preoperative Chemoradiotherapy for Rectal Cancer: Changes Over Time and Prognostic Significance. Technol Cancer Res Treat 2018, 17, 1533033818780065, doi:10.1177/1533033818780065.

Reviewer 2 Report

Jinming Fu et al. examined the relevance of blood count parameters as prognostic and predictive factors for overall (OS) and disease-free survival (DFS) of patients with colorectal cancers. They carried out a series of modelling experiments to develop prognostic inflammatory index for both OS and DFS. Generally, the paper is well planned and addresses a clinically relevant topic. However, the authors should take into consideration the following aspects that could influence the generalisability of their results:

  • Construction of OS-PII and DFS-PII is unclear. In ‘Methods’, the authors declare that both indices were developed using blood cell counts associated with OS and DFS based on relevant Cox models. However, Table S1 shows that lymphocytes and neutrophils were not significantly associated with OS and monocytes were not associated with DFS. Still, lymphocytes with neutrophils and monocytes were included in OS-PII and DFS-PII, respectively. The authors should explain why they included into their scores variables that were not independent predictors of survival as declared in ‘Methods’.
  • The authors should provide more details for their time-dependent ROC analyses (Table S8 and S9). Performance of the TNM variable is of particular interest as it showed best prediction for a single parameter. Did the authors used only two values (stage II and III) in their models and achieved AUCs of 0.65-0.72? This also applies to the combination of OS-PII with TNM.
  • Nomogram development should be explained in more detail, including selection procedure for the variables used. Since the final nomograms are relatively complex (more that 10 items), the authors should substantiate their application by comparing values of c-index for nomograms with some more clinically relevant systems (e.g. AJCC stages IIA, IIB, IIC, IIIA, IIIB, IIIC)
  • The authors should limit their modelling to 5 years since patient recruitment period ended in 2015.
  • Discussion should in greater detail evaluate the current nomogram with previously used systems. Moreover, the authors should comment how neoadjuvant treatment, routinely used for rectal cancer, could affect performance of the developed scores and nomograms.

Minor points:

  • Table 1 should include annotations for statistically significant differences between training and validation cohorts.
  • The authors should consider details of primary cancer sites (right colon, transverse colon, left colon, sigmoid colon, and rectum) to be used in table 1. Moreover, details for AJCC stages (IIA, IIB, IIC, IIIA, IIIB, IIIC) should be used instead of only stages II and III.
  • Survival curves (Fig. 2 and 3) should use appropriate time intervals on the x-axis (preferably years) as using 150 months generates some confusion (0 exposed patients)
  • Some references contain errors, i.e., no. 16 (authors and journal), ref. 29 (missing journal), ref. 44 (journal missing).
  • References for staining system used are missing
  • Figure S5 is analogous to fig. 3.

Author Response

Response to Reviewer 2 Comments

Point 1: Construction of OS-PII and DFS-PII is unclear. In ‘Methods’, the authors declare that both indices were developed using blood cell counts associated with OS and DFS based on relevant Cox models. However, Table S1 shows that lymphocytes and neutrophils were not significantly associated with OS and monocytes were not associated with DFS. Still, lymphocytes with neutrophils and monocytes were included in OS-PII and DFS-PII, respectively. The authors should explain why they included into their scores variables that were not independent predictors of survival as declared in ‘Methods’.

Response 1: Thank you for your valuable comment and question. We feel sorry that we did not describe the steps for PII construction clearly enough. In our study, the construction of PII is divided into three steps. First, restricted cubic spline regression was performed to determine whether there was a nonlinear relationship between six types of blood cell counts and survival (OS and DFS). In this step, platelet, lymphocyte, and eosinophil counts were found to have nonlinear relationships with OS, while platelet and eosinophil counts have nonlinear relationships with DFS. Thus, platelet, lymphocyte, and eosinophil counts were converted into binary variables using X-tile 3.6.1 software, and the others as continuous variables. Then, univariate Cox analyses of the above factors were performed respectively, in order to investigate the association between the six types of blood cell counts and OS or DFS. To avoid factors that were associated with the prognosis being excluded due to strict statistical p-value, blood cells whose statistical p<0.10 was defined as that associated with prognosis. According to the results reported in Table S1, platelet, lymphocyte, neutrophil, monocyte and eosinophil counts were associated with OS, whereas only platelet, neutrophil, monocyte and eosinophil counts were associated with DFS. Finally, significant prognostic factors (p<0.10) in the univariate analyses were entered into multivariate Cox models which were used to generate the weights of blood cell counts required to construct PII. The HRs and p-value obtained from multivariate Cox models were based on the mutual adjustment of different factors. Therefore, the results of the multivariate analysis could not be used to determine the association between blood count and prognosis. To make it clearer, we added annotations to Table S1 (see Table S1 in the supplementary document), which clearly described the result of Cox analyses and PII construction.

Point 2: The authors should provide more details for their time-dependent ROC analyses (Table S8 and S9). Performance of the TNM variable is of particular interest as it showed best prediction for a single parameter. Did the authors used only two values (stage II and III) in their models and achieved AUCs of 0.65-0.72? This also applies to the combination of OS-PII with TNM.

Response 2: Thank you for your insightful comments and suggestions. We followed your suggestion and have provided detailed annotations for Table S8-S11 (see page 10-13 in the supplementary document). As TNM staging system remains the gold standard for prognostication in CRC, the comparison of the prognostic accuracy (AUROCs) of TNM staging and PIIs, as well as the comparison of the prognostic accuracy (AUROCs) of TNM staging and combinations of TNM staging and PIIs is of particular interest in our study. To evaluate the AUROCs of TNM staging, we used two values (stage II and III) in their models and achieved AUCs of 0.65-0.72 at different time points. This also applies to the combination of OS-PII with TNM.

Point 3: Nomogram development should be explained in more detail, including selection procedure for the variables used. Since the final nomograms are relatively complex (more that 10 items), the authors should substantiate their application by comparing values of c-index for nomograms with some more clinically relevant systems (e.g. AJCC stages IIA, IIB, IIC, IIIA, IIIB, IIIC)

Response 3: Thank you for your kind and valuable advice. We have revised the Materials and Methods section and explained the nomogram development in more detail. We also described the selection criteria for prognostic factors incorporated into the nomograms, that is “Multivariate Cox analyses determined the effects of prognostic factors on a nomogram, and only the factors with a p-value < 0.05 were finally incorporated into the nomogram” (see line 415-416 on page 17 in the manuscript). Followed by the reviewer's suggestion, we evaluated the c-index of nomograms and AJCC system for predicting both overall survival and disease-free survival and found that compared with AJCC system, nomograms had higher C-index (see Table S12 in the supplementary document). We also performed the decision curve analyses for the prognostic models of nomograms and AJCC system (see Figure S9-10 in the supplementary document). Decision curve analyses for the prognostic models of the AJCC system and nomograms indicated that both two prognostic models showed a positive net benefit in predicting 5-year OS and DFS in the training and validation cohorts (Figure S9-10). Compared with the AJCC system, the nomograms have better clinical applicability because of their wider range of threshold probabilities and higher net benefit (Figure S9-10). The developed nomograms were worth using in terms of personalized survival prediction of patients with CRC (see line 245-250 on page 12).

Table S12. The C-index of AJCC system and nomograms.

Models

Training cohort (N=4154)

Validation cohort (N=5161)

C-index

95% CI

C-index

95% CI

Overall Survival

AJCC system

0.654

0.639-0.669

0.700

0.678-0.722

Nomogram

0.718

0.704-0.731

0.765

0.745-0.785

Disease-free Survival

AJCC system

0.654

0.638-0.671

0.657

0.640-0.674

Nomogram

0.700

0.684-0.716

0.698

0.681-0.715

Point 4: The authors should limit their modelling to 5 years since patient recruitment period ended in 2015.

Response 4: Thank you for your comment. In our study, data collection for both two cohorts began in 2007, but the end time of follow-up was different. The training cohort included 4154 patients obtained from the Third Affiliated Hospital of Harbin Medical University between January 2007 and December 2013, and the validation cohort included 5161 patients obtained between January 2007 and December 2015 from the Fudan University Shanghai Cancer Center. Therefore, for the validation cohort, the follow-up time of a few patients can't reach 5 years. Although the median follow-up time was 70.0 months (interquartile ranges: 50.0-94.0) in the training cohort, the follow-up time among different patients still varies greatly. Therefore, we chose to construct the PIIs and perform Cox models based on the whole follow-up. Both OS-PII and DFS-PII had a stable prognostic performance at various follow-up times according to the time-dependent receiver operating characteristic analyses, which also implied the stability of the PIIs.

Point 5: Discussion should in greater detail evaluate the current nomogram with previously used systems. Moreover, the authors should comment how neoadjuvant treatment, routinely used for rectal cancer, could affect performance of the developed scores and nomograms.

Response 5: Thank you for your suggestion. We evaluated the c-index of nomograms and AJCC system, in terms of both overall survival and disease-free survival prediction (see Table S12 in the supplementary document), and we also performed the decision curve analyses for the prognostic models of nomograms and AJCC system, to evaluate the clinical applicability of these two models (see line 245-250 on page 12 in the manuscript). In addition, we discussed this part detailedly “AJCC system is considered as the benchmark for classifying patients with cancer and defining prognosis. Compared with AJCC staging, the nomograms had higher C-index and net benefit, which implies the better clinical applicability of these nomograms” in the Discussion (see line 338-341 on page 16 in the manuscript). As the patient's blood cell counts could be affected by preoperative neoadjuvant treatment, we excluded the patients who took neoadjuvant treatment before surgery. Therefore, we could not comment on how neoadjuvant treatment, routinely used for rectal cancer, could affect the performance of the developed scores and nomograms. Considering whether postoperative chemotherapy would affect the prognosis of the PIIs, we did subgroup analyses stratified by postoperative chemotherapy (no or yes), and the results showed that the prognostic effects of OS-PII and DFS-PII between the postoperative chemotherapy group and the non-chemotherapy group were not significantly different (see Figure S6-S7 in the supplementary document).

Point 6: Minor points:

(1) Table 1 should include annotations for statistically significant differences between training and validation cohorts.

Response: Thank you for your valuable suggestion. We all glad to follow your suggestion, and have performed Student’s t tests for normally distributed continuous variables, χ2 tests for categorical variables, Mann-Whitney U tests for non-normally distributed continuous variables, to evaluate the differences between training and validation cohorts (see line 401-403 on page 17 in the manuscript). We also added p values in Table 1 to describe the statistically differences between training and validation cohorts (see Table 1).

(2) The authors should consider details of primary cancer sites (right colon, transverse colon, left colon, sigmoid colon, and rectum) to be used in table 1. Moreover, details for AJCC stages (IIA, IIB, IIC, IIIA, IIIB, IIIC) should be used instead of only stages II and III.

Response: Thank you for your careful suggestion. We have revised Table 1 that described detailedly the primary cancer sites (right colon, transverse colon, left colon, sigmoid colon, and rectum) and AJCC stages (IIA, IIB, IIC, IIIA, IIIB, IIIC). Your suggestion made our presentation to be clearer.

(3) Survival curves (Fig. 2 and 3) should use appropriate time intervals on the x-axis (preferably years) as using 150 months generates some confusion (0 exposed patients)

Response: Thank you for your suggestion. We feel very sorry that the time intervals of Figure 2 and Figure 3 were inappropriate and may generates some confusion. To make our manuscript more rigorous, we revised Figure 1, Figure 2 and Figure S5, and used years as the time interval on the x-axis (see Figure 1and Figure 2 in the manuscript; see Figure S5 in the supplementary document). The longest survival time of patients was 12 year and 8 year in the training and validation cohort, respectively.

(4) Some references contain errors, i.e., no. 16 (authors and journal), ref. 29 (missing journal), ref. 44 (journal missing).

Response: Thank you for your careful suggestion. We apologize for our carelessness. We carefully examined all the references in the manuscript and found that ref. 16, ref. 29, ref. 40 and ref. 44 contain errors. We have corrected these mistakes and checked them carefully one more time. Thank you for your kind suggestion once more.

(5) References for staining system used are missing.

Response: Thank you for your suggestion. We feel sorry that we did not describe the measurement technology of blood routine tests in the Materials and Methods section. In our study, peripheral blood cell counts were measured by an autoanalyzer (Sysmex XE-2100, Kobe, Japan) and did not require staining system. We have added the description of the measurement technology of blood routine tests in the revised version that could make our manuscript clearer (see line 374-378 on page 17 in the manuscript).

(6) Figure S5 is analogous to fig. 3.

Response: Thank you for your careful reminder. We agree with you that Figure S5 is analogous to Figure 3. Therefore, we deleted Figure S5 to make our study more concise.

Reviewer 3 Report

The authors designed in their retrospective correlative study an inflammatory prognostic index for colon and rectal cancer patients stage II and III. They established the index using readily available blood values in a cohort from one hospital and validated the results in a second cohort from an other hospital.

This is a very important study for colon and rectal cancer patients stage II and III in order to further determine patients with high risk of recurrence or shorter survival to target such Groups in the future with adjuvant chemotherapy or intensified adjuvant chemotherapy. The Groups are fairly large with  4154 patients and 5161 patients.

Minor Points:

Change the abbreviation of the risk groups from G1-4 to RG 1-4. G is the abbreviaton of grading and readers may get confused.

Author Response

Response to Reviewer 3 Comments

Point 1: Change the abbreviation of the risk groups from G1-4 to RG 1-4. G is the abbreviaton of grading and readers may get confused.

Response 1: Thank you very much for your kind suggestion. We have changed the abbreviation of the risk groups from G1-4 to RG 1-4 (see line 186-196 on page 10 and Figure 2 in the manuscript; see Figure S5 in the supplementary document), and have defined RG as “risk group” in parentheses the first time it appears in the main text (see line 186-196 on page 10 in the manuscript). This suggestion helped us to clarity of this manuscript. Thank you very much once more.

Round 2

Reviewer 2 Report

The authors have solved all major questions raised by the previous manuscript.